# Comparative Genomics of *Mortierellaceae* Provides Insights into Lipid Metabolism: Two Novel Types of Fatty Acid Synthase

**DOI:** 10.3390/jof8090891

**Published:** 2022-08-23

**Authors:** Heng Zhao, Yong Nie, Yang Jiang, Shi Wang, Tian-Yu Zhang, Xiao-Yong Liu

**Affiliations:** 1College of Life Sciences, Shandong Normal University, Jinan 250358, China; 2Institute of Microbiology, School of Ecology and Nature Conservation, Beijing Forestry University, Beijing 100083, China; 3School of Civil Engineering and Architecture, Anhui University of Technology, Ma’anshan 243002, China; 4State Key Laboratory of Mycology, Institute of Microbiology, Chinese Academy of Sciences, Beijing 100101, China

**Keywords:** Mucoromyceta, microbial lipids, phylogenomics, polyunsaturated fatty acids, arachidonic acid

## Abstract

Fungal species in the family *Mortierellaceae* are important for their remarkable capability to synthesize large amounts of polyunsaturated fatty acids, especially arachidonic acid (ARA). Although many genomes have been published, the quality of these data is not satisfactory, resulting in an incomplete understanding of the lipid pathway in *Mortierellaceae*. We provide herein two novel and high-quality genomes with 55.32% of syntenic gene pairs for *Mortierella alpina* CGMCC 20262 and *M. schmuckeri* CGMCC 20261, spanning 28 scaffolds of 40.22 Mb and 25 scaffolds of 49.24 Mb, respectively. The relative smaller genome for the former is due to fewer protein-coding gene models (11,761 vs. 13,051). The former yields 45.57% of ARA in total fatty acids, while the latter 6.95%. The accumulation of ARA is speculated to be associated with delta-5 desaturase (Delta5) and elongation of very long chain fatty acids protein 3 (ELOVL3). A further genomic comparison of 19 strains in 10 species in three genera in the *Mortierellaceae* reveals three types of fatty acid synthase (FAS), two of which are new to science. The most common type I exists in 16 strains of eight species of three genera, and was discovered previously and consists of a single unit with eight active sites. The newly revealed type II exists only in *M. antarctica* KOD 1030 where the unit is separated into two subunits α and β comprised of three and five active sites, respectively. Another newly revealed type III exists in *M. alpina* AD071 and *Dissophora globulifera* REB-010B, similar to type II but different in having one more acyl carrier protein domain in the α subunit. This study provides novel insights into the enzymes related to the lipid metabolism, especially the ARA-related Delta5, ELOVL3, and FAS, laying a foundation for genetic engineering of *Mortierellaceae* to modulate yield in polyunsaturated fatty acids.

## 1. Introduction

Environmental protection and sustainable development are now giving urgency to identifying alternatives for oil plants and animals. Therefore, oleaginous fungi such as *Aspergillus flavus* Link [=*A. oryzae* (Ahlb.) Cohn], *Mortierella alpina* Peyronel, *M. wolfii* B.S. Mehrotra & Baijal, *Mucor racemosus* Fresen., *Lichtheimia corymbifera* (Cohn) Vuill., *Syncephalastrum racemosum* Cohn ex J. Schröt., *Geotrichum candidum* Link, *Umbelopsis isabellina* (Oudem.) W. Gams (=*Mortierella isabellina* Oudem.), and *Yarrowia lipolytica* (Wick., Kurtzman & Herman) Van der Walt & Arx have recently gained attention and played an important role in the studies of biodiesel and polyunsaturated fatty acids (PUFAs) [1,2,3,4,5,6,7,8,9,10,11]. These oleaginous fungi mainly belong to *Ascomycota*, *Mortierellomycota*, and *Mucoromycota*.

Compared with other microorganisms such as microalgae and algae, oleaginous fungi are distinct in possessing a shorter life cycle, assimilating a variety of carbon sources, and adapting to various weather and seasons [1,7,9]. Meanwhile, the species in the *Mortierellaceae* of *Mortierellomycota* are characterized by synthesizing a large amount of PUFAs [1,12]. At present, 122 species of seven genera are accommodated in the family *Mortierellaceae* [13], and almost all tested strains of *Mortierellaceae* were found to be able to synthesize arachidonic acid (ARA, C20:4) [1,12,14,15,16], which is a vital nutrient for the elderly and infants, possessing multiple functions such as the protection of brain and muscles and potential against tumors and inflammation [16,17,18].

*Mortierella alpina* synthesizing lipids more than 50% of its dry cell weight and ARA accounting for 30–70% of the total fatty acids has recently been one of the important model organisms for PUFA metabolism [12,16,19,20,21,22]. Based on genetic manipulation in *M. alpina*, numerous enzymes such as AMP deaminase, omega-3 desaturase, delta-5 desaturase, and fatty acid synthase were proved to play key functions in the biosynthesis of PUFAs [12,16,23,24,25]. For example, overexpression of homologous AMP deaminase in *M. alpina* resulted in a significant increase of 15.0–34.3% in lipid yield [23].

In 2011, the first draft genome of *Mortierella alpina* was completed for the strain ATCC 32222, initializing studies on lipid metabolic pathway in the species [11]. Until now, nine genomes of *M. alpina* have been published in the NCBI database (https://www.ncbi.nlm.nih.gov, accessed on 2 July 2022) [12,26,27,28]. Other genomes of the *Mortierellaceae* were assembled for *Actinomortierella wolfii* (B.S. Mehrotra & Baijal) Vandepol & Bonito, *Dissophora globulifera* (O. Rostr.) Vandepol & Bonito, *Mortierella amoeboidea* W. Gams, *M. antarctica* Linnem., *M. elongata* Linnem., *M. epicladia* W. Gams & Emden, *M. gamsii* Milko, and *M. verticillata* Linnem. [27,29,30,31]. However, comprehensive analyses of these genomes, especially the key genes involved in fatty acid synthesis, are not satisfactory due to a lack of good-quality sequence data.

In the present study, we sequence two strains, *Mortierella alpina* CGMCC 20262 and *M. schmuckeri* Linnem. CGMCC 20261, by combining next-generation with PacBio sequencing. A total of 40 strains of basal fungi are used to reconstruct their phylogenomic relationship, and 13 protein-coding genes involved in the lipid metabolism are characterized in 19 strains of *Mortierellaceae*.

## 2. Materials and Methods

### 2.1. Strains, Media, and Fermentation

*Mortierella alpina* CGMCC 20262 and *M. schmuckeri* CGMCC 20261 were deposited in the China General Microbiological Culture Collection Center, Beijing, China (CGMCC). Cultures were incubated with potato dextrose agar (PDA: 200 g/L potato, 20 g/L glucose, 20 g/L agar, and 1000 mL distilled water) and ampicillin (100 μg/mL) at 20 °C for ten days. A shake flask fermentation was then carried out following a previous study [1]. In brief, 1 mL of spore suspension (1 × 10^6^) was incubated at 20 °C and 140 rpm for seven days in a 250 mL flask with 100 mL of modified Kendrick media (50 g/L glucose, 2 g/L diammonium tartrate, 7 g/L KH_2_PO_4_, 2 g/L Na_2_HPO_4_, 1.5 g/L MgSO_4_·7H_2_O, 1.5 g/L yeast extract, 0.1 g/L CaCl_2_·2H_2_O, 8 mg/L FeCl_3_·6H_2_O, 1 mg/L ZnSO_4_·7H_2_O, 0.1 mg/L CuSO_4_·5H_2_O, 0.1 mg/L CO(NO_3_)_2_·6H_2_O, 0.1 mg/L MnSO_4_·5H_2_O, and pH 6.0) [1,23,32].

### 2.2. Biomass, Fatty Acid Measurement, and Profiling

Biomass, fatty acid measurement, and profiling followed the method by Zhao et al. [1]. Fresh biomasses were drained with a vacuum pump, and then freeze-dried for two days to attain a constant dry cell weight (DCW). The freeze-dried biomasses were hydrolyzed with HCl solution (6 mol/L), and then extracted with ethanol, anhydrous ether, and petroleum ether following previous studies [1,33]. The extract was then dried with an oven by slowly warming to 80 °C until a constant total lipid weight (TLW) was attained. Total lipid content (TLC) was calculated as the TLW being divided by the DCW. For fatty acid profiling, the gas chromatography–mass spectrometry (GC/MS, QP2010, Shimadzu Corp., Japan) was performed with a 30 m × 0.25 mm × 0.25 μm column (Rtx-5MS, RESTEK, Bellefonte, PA, USA), and nonanoic acid (C9:0) was selected as internal standard according to Zhao et al. [1].

### 2.3. Genome Sequencing and Assembly

*Mortierella alpina* CGMCC 20262 and *M. schmuckeri* CGMCC 20261 were incubated with PDA at 20 °C for five days. Total cell DNAs were extracted from mycelia using a kit (O-GPLF-400, GeneOnBio Corporation, Changchun, China) according to the manufacturer’s protocol, and then detected by DNA/Protein Analyzer and 1% agarose gel electrophoresis. High-quality DNAs were sequenced at Beijing Novogene Bioinformatics Technology Co., Ltd. (Beijing, China) by using a PacBio Sequel and Illumina NovaSeq 6000 platform with 20 kb and 350 bp library, respectively. Low-quality reads (less than 500 bp) were removed by quality control from the raw data produced with the PacBio Sequel platform, and then the controlled high-quality reads were de novo assembled using SMRT Link v5.1.0 [34]. The assembled genomes were assessed by Quast v5.0.2 [35] and BUSCO v5.2.2 [36].

### 2.4. Gene Prediction and Functional Annotation

Protein-coding gene models of *Mortierella alpina* CGMCC 20262 and *M. schmuckeri* CGMCC 20261 were de novo predicted using Augustus v3.3.3 [37]. Amino acid and DNA sequences were functionally annotated using NR (https://www.ncbi.nlm.nih.gov/protein/, accessed on 2 July 2022), NT (https://www.ncbi.nlm.nih.gov/nucleotide/, accessed on 2 July 2022), Pfam [38], GO [39], KEGG [40], CAZymes [41,42], and Eggnog [43] databases. All sequences were mapped onto these databases using Diamond v2.0.1 [44] with an e-value less than 1 × 10^−5^. The antiSMASH fungal version [45] was used to annotate the gene clusters of secondary metabolites with default parameters. Repetitive elements were identified using the Extensive de novo TE Annotator (EDTA) pipeline v1.9.5 [46]. RNAmmer v1.2 [47] and tRNAscan-SE v2.0.5 [48] were used to predict rRNAs and tRNAs, respectively.

### 2.5. Phylogenomic and Phylogenetic Analyses

A total of 40 strains of early diverging fungi, including 38 strains downloaded from online databases and two strains sequenced herein, were used for phylogenomic analyses (Table 1). A total of 192 clusters of orthologous proteins were identified with HMMER v3.3.1 [49] and Trimal v1.4.4 [50], following the methods described by Spatafora et al. [51] and James et al. [52], and then their amino acid sequences were aligned with MAFFT v7 [53]. Phylogenomic analyses were carried out with a Maximum Likelihood (ML) algorithm using RaxML v8.1.12 [54]. Maximum Likelihood analyses adopted the PROTGAMMALGX substitution model with 100 bootstrap replications. Phylogenetic analyses on amino acids of delta-5 desaturase and fatty acid synthase were performed using IQ-TREE v1.0 [55] with an ML algorithm, WAG substitution model, and 1000 bootstrap replications.

### 2.6. Comparative Genomic Analyses

The genome sequences of *M**ortierella alpina* CGMCC 20262 and *M. schmuckeri* CGMCC 20261 were aligned using MCScanX for all protein-coding gene models, and then genomic collinearity was analyzed using a dual synteny plotter package [61]. For comparison of genes in lipid metabolism, especially those encoding fatty acid synthase and delta-5 desaturase, all 19 genomes of *Mortierellaceae* were reannotated using the KEGG database [41].

## 3. Results

### 3.1. Fatty Acid Profiles of Mortierella alpina and M. schmuckeri

*Mortierella alpina* CGMCC 20262 is much lower than *M. schmuckeri* CGMCC 20261 in dry cell weight (DCW, 6.6 g/L vs. 11.7 g/L), total lipid weight (TLW, 1.1 g/L vs. 8.0 g/L), and total lipid content (TLC, 17% vs. 70%; Figure 1a). However, *M. alpina* CGMCC 20262 yields 75.54% of polyunsaturated fatty acids (PUFAs; 15.41% of C18:2, 8.78% of C18:3, 3.40% of C20:3, 45.57% of C20:4, and 2.39% of C20:5) in total fatty acids (Figure 1b), more than the percentage of 23.03% in *M. schmuckeri* CGMCC 20261 (8.30% C18:2, 4.36% of C18:3, 3.43% of C20:3, 6.95% of C20:4, and no C20:5). Obviously, these two strains are significantly different, C20:4 (45.57%) being the top one in *M. alpina* CGMCC 20262, and C18:1 (27.54%) and C16:0 (24.68%) being the top two in *M. schmuckeri* CGMCC 20261 (Figure 1b).

### 3.2. Genomic Features of Mortierella alpina and M. schmuckeri

Assembled genomes span 28 scaffolds of 40.22 Mb with GC content of 50.92% in *Mortierella alpina* CGMCC 20262, and 25 scaffolds of 49.24 Mb with GC content of 47.46% in *M. schmuckeri* CGMCC 20261 (Table 2). The number of predicted protein-coding gene models of *M. schmuckeri* CGMCC 20261 is slightly more than that of *M. alpina* CGMCC 20262 (13,051 vs. 11,761; Table 2). Among these gene models, 66.17%, 33.97%, 47.35%, 42.69%, 73.20%, 37.71%, and 1.70% are mapped onto Pfam, NT, NR, GO, Eggnog, KEGG, and CAZymes databases in *M. schmuckeri* CGMCC 20261, and 72.33%, 34.23%, 41.70%, 46.01%, 77.17%, 41.31%, and 1.95% in *M. alpina* CGMCC 20262 (Table 2). Repetitive elements of *M. alpina* CGMCC 20262 and *M. schmuckeri* CGMCC 20261 account for 7.29% and 7.45% of the whole genomes (Table 2). Furthermore, 38 rRNAs (11 of 8s rRNA, 12 of 18s rRNA, and 15 of 28s rRNA) and 29 rRNAs (9 of 8s rRNA, 8 of 18s rRNA, and 12 of 28s rRNA) are predicted in *M. alpina* CGMCC 20262 and *M. schmuckeri* CGMCC 20261, respectively. More characteristics of genomes are listed in Table 2.

### 3.3. Phylogenomic Placements of Mortierella alpina and M. schmuckeri

The Maximum Likelihood phylogenomic tree (Figure 2) suggests that *Mortierella alpina* CGMCC 20262 is closely related to other strains of *M. alpina* (Maximum Likelihood bootstrap values, MLBV = 100%), and *M. schmuckeri* CGMCC 20261 is a sister to *M. gamsii* (MLBV = 100%) and closely related to *M. elongata* (MLBV = 100%).

### 3.4. Synteny between Mortierella alpina and M. schmuckeri

A total of 13,726 gene models located in all scaffolds are collinear between *Mortierella alpina* CGMCC 20262 and *M. schmuckeri* CGMCC 20261, accounting for 55.32% of all the 24,812 gene models in both strains. Figure 3 shows the synteny of the ten largest scaffolds where 16,927 gene models are predicted and 11,777 (69.58%) gene models are syntenic.

### 3.5. Lipid Metabolism in Mortierellaceae

To explore the lipid metabolism in *Mortierellaceae*, genomes of 19 strains in 10 species are de novo annotated or reannotated. We focus on the 13 genes associated with NADPH, precursors, desaturation, and elongation (Figure 4 and Appendix A). The gene *Delta5* (encoding delta-5 desaturase) is not found in *M. gamsii* NVP60, and neither is *Delta6* (encoding delta-6 desaturase) in *M. alpina* LL118. With these two exceptions, each gene possesses between one and six copies. The genes *HK* (encoding hexokinase, a key enzyme in EMP or the glycolytic pathway) and *ME* (encoding malate dehydrogenase) are remarkable as both have between three and six copies (Appendix A).

Three types of fatty acid synthase are identified in the family *Mortierellaceae* (Figure 5, Appendix A). Type I is ubiquitous (*Actinomortierella wolfii* NRRL 6351, *Mortierella alpina* AD072, *M. alpina* ATCC 32222, *M. alpina* B6842, *M. alpina* CCTCC M-207067, *M. alpina* CGMCC 20262, *M. alpina* CK1249, *M. alpina* GBA31, *M. alpina* LL118, *M. alpina* NRRL 66262, *M. amoeboidea* CBS 889.72, *M. elongata* AG-77, *M. epicladia* AD058, *M. schmuckeri* CGMCC 20261, and *M. verticillata* NRRL 6337), and unites together eight active sites (Figure 5). Type II is found in *M. antarctica* KOD 1030 only and includes two subunits, α subunit involving KR, KS, and PPT active sites, and β subunit comprising AT, ER, DH, MPT, and ACP active sites. Type III is identified in *Dissophora globulifera* REB-010B and *Mortierella alpina* AD071, similar to type II but with one more ACP active site in α subunit.

Four desaturases, i.e., *Delta5*, *Delta6*, *Delta9*, and *Delta12*, are found in the 19 strains of *Mortierellaceae* (Appendix A). Among these, the *Delta5* catalyzes the synthesis of arachidonic acid (ARA, C20:4; Figure 4), which has diverged into three clades in *Mortierellaceae* (Figure 6 and Appendix A).

The elongation of very long chain fatty acids protein (ELOVL) family consists of four members in *Mortierellaceae*, i.e., *ELOVL2*, *ELOVL3*, *ELOVL4*, and *ELOVL6* (Appendix A). *ELOVL2* is shared by all strains of *Mortierellaceae*, while *ELOVL3*, *ELOVL4*, and *ELOVL6* are species-specific. In detail, *ELOVL3* is identified in *M. alpina* and *M. amoeboidea*, and *ELOVL4* and *ELOVL6* are annotated in *M. verticillata* and *D. globulifera*.

## 4. Discussion

Among numerous oleaginous fungi, *Mortierella alpina* is now attracting more and more attention because of its ability to accumulate large amounts of arachidonic acid (ARA) [12,16,22]. Previous studies suggested that *M. alpina* synthesized lipids up to 50% of dry cell weight (DCW) and ARA 30–70% of total lipid content (TLC) [12]. Herein, *M. alpina* CGMCC 20262 falls in these ranges, yielding lipids 17% of DCW and ARA 45.57% of TCL (Figure 1). In addition, we find a novel oleaginous fungus, *M. schmuckeri* CGMCC 20261, which accumulates lipids 70% of DCW (Figure 1), much more than *M. alpina* (17%).

The draft genome of *M. alpina* was published in 2011, and consequently lipid synthesis pathway was predicted [12]. Since then, more genomes of members in the family *Mortierellaceae* have shed light on their lipid synthesis mechanism. For example, *Mortierella* sp. BCC40632 has an n-6 series fatty acid synthesis pathway due to a lack of delta-15 or omega-3 desaturase [25,26,27,30,31]. In this paper, the genomes of two more strains, *M. alpina* CGMCC 20262 and *M. schmuckeri* CGMCC 20261, are sequenced to provide more insights into lipid metabolism (Table 2).

A large number of genes hitherto have been found to affect lipid synthesis in *Mortierella alpina*, such as acetyl-CoA carboxylase, AMP deaminase, malate dehydrogenase, delta-5 desaturase, delta-6 desaturase, and elongase 2 [12,22,23,66,67,68,69]. In this study, we compare 13 genes of 19 strains in *Mortierellaceae* (Appendix A), finding differences in genes encoding hexokinase, delta-5 desaturase, and elongation of very long chain fatty acids protein. Compared with *M. schmuckeri* CGMCC 20261, the *M. alpina* CGMCC 20262 possesses one more copy of the *HK* gene, more copies of *Delta5* and *ELOVL* genes but fewer copies of the *Delta9* gene (Appendix A). Delta-5 desaturase and elongation of very long chain fatty acids protein play an essential role in the synthesis of polyunsaturated fatty acids (PUFAs, Figure 4), which might be one of the reasons for *M. alpina* CGMCC 20262 yielding more PUFAs than *M. schmuckeri* CGMCC 20261.

PUFA biosynthesis requires an activity of elongation of very long chain fatty acids proteins, as well as fatty acid desaturases such as delta-5, delta-6, delta-12, and omega-3 desaturase [12,16]. For example, the *M. alpina* 1S-4 mutant in delta-5 desaturase resulted in a huge accumulation of dihomo-γ-linolenic acid (DGLA), up to 43.3% of total fatty acids [70]. In the present study, the delta-5 desaturase of *M. alpina* is significantly different from other species in the family *Mortierellaceae* based on the phylogenetic analyses (Figure 6 and Appendix A), which is probably one of the reasons for synthesizing high content of ARA [12,71,72]. Elongation of very long chain fatty acids proteins (ELOVLs) are involved in the long-chain polyunsaturated fatty acid synthetic pathway, and four gene types, *ELOVL2*, *ELOVL3*, *ELOVL4*, and *ELOVL6*, are identified herein in 19 strains of *Mortierellaceae* (Appendix A).

The fatty acid synthase plays an important role in lipid synthesis, catalyzing the synthesis of saturated fatty acids from acetyl-CoA and malonyl-CoA. A variety of fatty acid synthases were found in fungi, such as *Aspergillus oryzae*, *Mortierella alpina*, *Saccharomyces cerevisiae*, and *Yarrowia lipolytica* [6,12,73,74]. In this study, three types of fatty acid synthases are found in *Mortierellaceae* (Appendix A). Type I is the most common and similar to that reported in *Mortierella alpina* ATCC 32222, a single subunit consisting of eight catalytic domains/active sites [17]. Types II and III are new to science. Type II presents in *M. antarctica* KOD 1030 only, and type III in *Dissophora globulifera* REB-010B and *Mortierella alpina* AD071 (Appendix A).

Overall, the rapid increase in genomic data is providing more materials for tackling fatty acid biosynthesis, discovering more potential genes, and finalizing the understanding of the genetic foundation of lipid metabolism.

## 5. Conclusions

In this paper, two new genomes, *Mortierella alpina* CGMCC 20262 and *M. schmuckeri* CGMCC 20261, were sequenced using the PacBio Sequel and Illumina NovaSeq 6000 platform. To explore the lipid metabolism in *Mortierellaceae*, a total of 19 genomes were reannotated. The results suggest that delta-5 desaturase and elongation of very long chain fatty acids protein 3 probably promoted the accumulation of polyunsaturated fatty acids, especially arachidonic acid. Besides, three types of fatty acid synthase, including two novel ones, were identified. Consequently, with the increase in public genomic data, a comprehensive analysis on lipid metabolism will discover more genes or protein-encoding models, providing more information for subsequent genetic engineering of oleaginous fungi.

## Figures and Tables

**Figure 1 jof-08-00891-f001:**
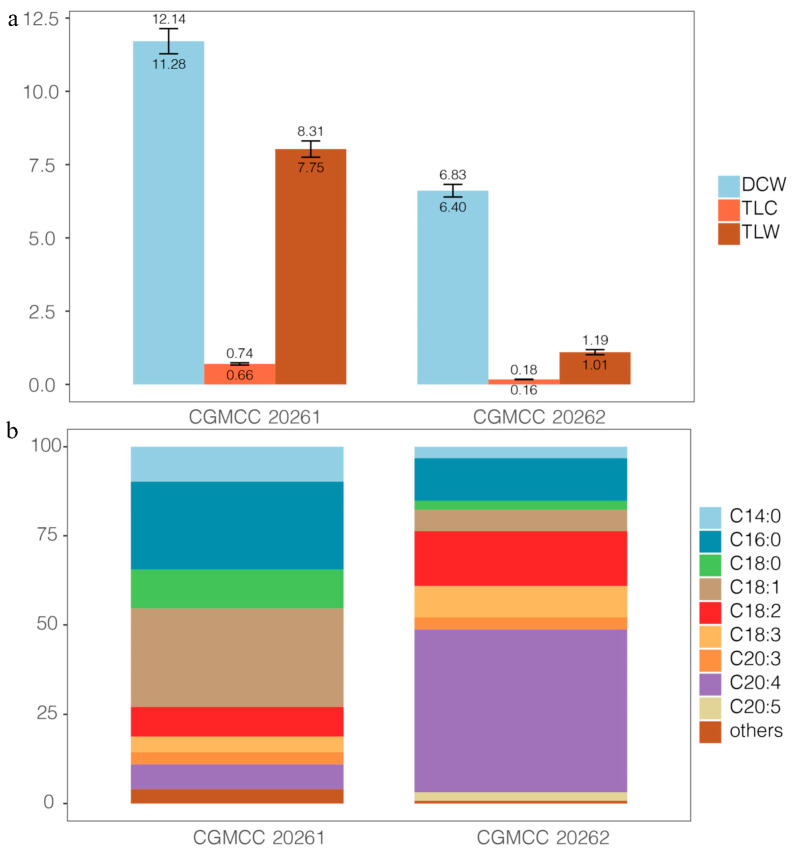
Lipids of *Mortierella alpina* CGMCC 20262 and *M. schmuckeri* CGMCC 20261. (**a**) Lipid characters: dry cell weight (DCW, g/L), total lipid weight (TLW, g/L), and total lipid content (TLC); (**b**) fatty acid profiles: others include C10:0, C12:0, C15:0, C20:0, and C20:1.

**Figure 2 jof-08-00891-f002:**
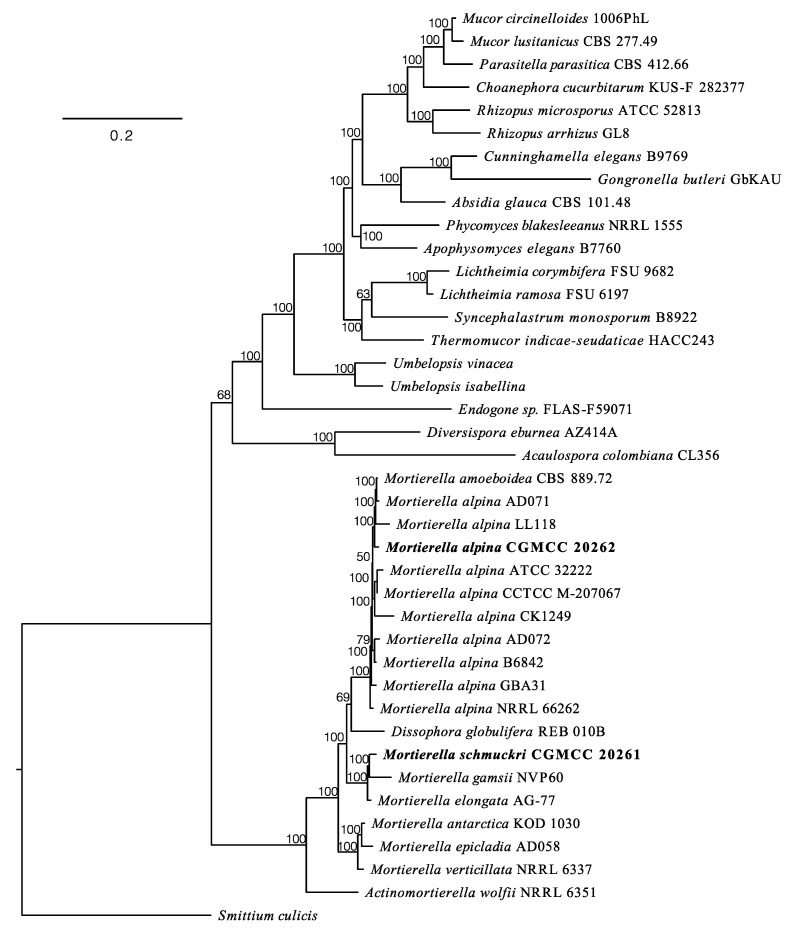
A Maximum Likelihood phylogenomic tree illustrating the placements of *Mortierella alpina* CGMCC 20262 and *M. schmuckeri* CGMCC 20261 based on 192 clusters of orthologous proteins. New genomes obtained in this study are in bold. Maximum Likelihood bootstrap values (MLBV ≥ 50%) are indicated along branches. A scale bar in the upper left indicates substitutions per site.

**Figure 3 jof-08-00891-f003:**
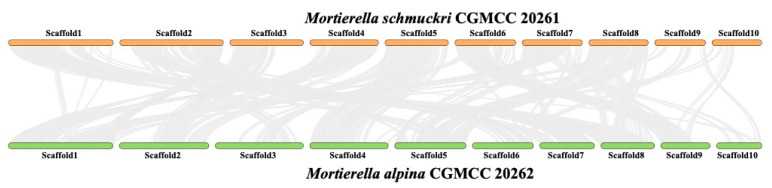
Genomic synteny of *Mortierella alpina* CGMCC 20262 and *M. schmuckeri* CGMCC 20261 based on protein-coding gene models of the 10 largest scaffolds.

**Figure 4 jof-08-00891-f004:**
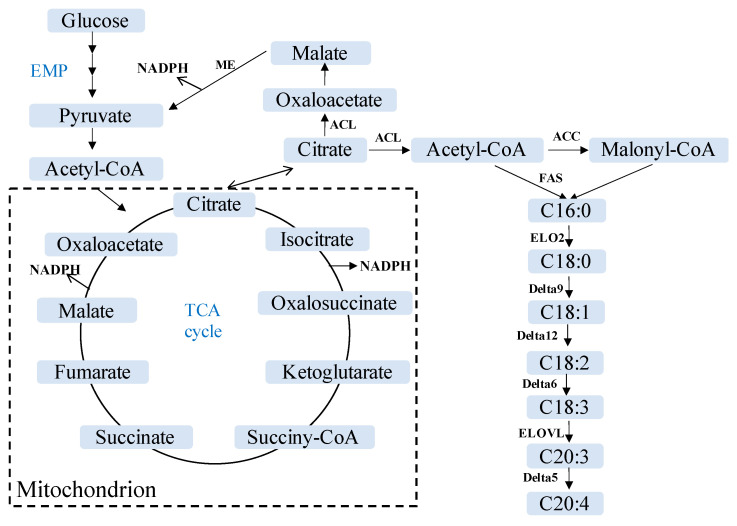
The polyunsaturated fatty acid synthesis mechanism in *Mortierellaceae*. EMP: Glycolytic pathway; TCA cycle: Tricarboxylic acid cycle; ME: Malate dehydrogenase; NADPH: Nicotinamide adenine dinucleotide phosphate; ACC: Acetyl-CoA carboxylase; ACL: ATP citrate (pro-S)-lyase; FAS: Fatty acid synthase; ELO2: Fatty acid elongase 2; Delta5: delta-5 desaturase; Delta6: delta-6 desaturase; Delta9: delta-9 desaturase; Delta12: delta-12 desaturase; ELOVL: Elongation of very long chain fatty acids protein.

**Figure 5 jof-08-00891-f005:**
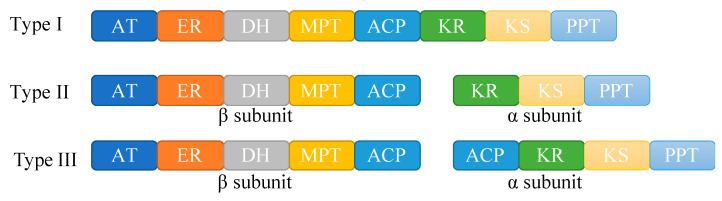
Fatty acid synthases in *Mortierellaceae*. AT: Acyl transferase domain; ER: Enoyl reductase domain; DH: dehydratase; MPT: Malonyl CoA-acyl carrier protein transacylase; ACP: Acyl carrier protein domain; KR: β-ketoacyl reductase; KS: β-ketoacyl synthase; PPT: Phosphopantetheinyl transferase.

**Figure 6 jof-08-00891-f006:**
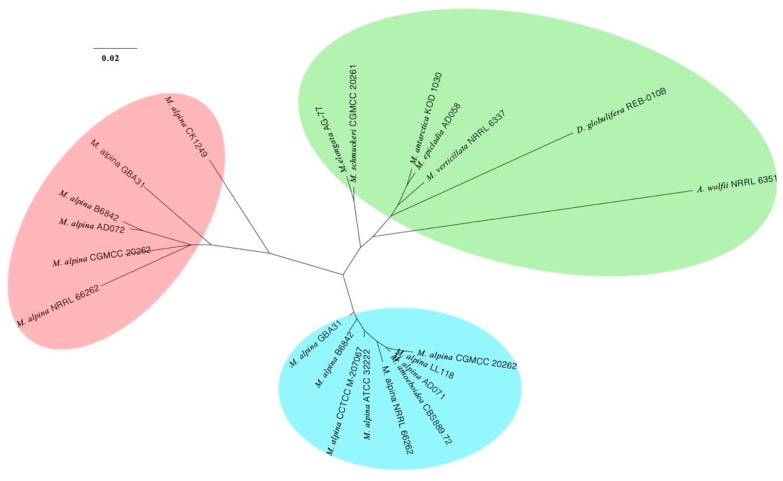
A Maximum Likelihood phylogenetic tree of the gene encoding delta-5 desaturase in the family *Mortierellaceae*. A scale bar in the upper left indicates substitutions per site.

**Table 1 jof-08-00891-t001:** The genomic information used for phylogenomic analyses in this study.

Species	Strains	BioSample	References
*Absidia glauca*	CBS 101.48	SAMEA3923633	[56]
*Acaulospora morrowiae*	CL551	SAMEA8911292	
*Apophysomyces elegans*	B7760	SAMN02351510	
*Actinomortierella wolfii **	NRRL 6351	SAMN05720777	[27]
*Choanephora cucurbitarum*	KUS-F282377	SAMN04532838	
*Cunninghamella elegans*	B9769	SAMN02351511	
*Dissophora globulifera **	*REB 010B*	SAMN05720531	[27]
*Diversispora eburnea*	AZ414A	SAMEA8911293	
*Endogone* sp.	FLAS-F59071	SAMN09071421	[57]
*Gongronella butleri*	GbKAU	SAMN15221701	
*Lichtheimia corymbifera*	FSU 9682	SAMEA2189700	[58]
*L. ramosa*	FSU 6197	SAMN05179542	[59]
** *Mortierella alpina ** **	**CGMCC 20262**	**SAMN29490473**	**This study**
*M. alpina **	AD071	SAMN05720461	[27]
*M. alpina **	AD072	SAMN05720462	[27]
*M. alpina **	ATCC 32222	SAMN02981246	[12]
*M. alpina **	B6842	SAMN02370960	[26]
*M. alpina **	CCTCC M-207067	SAMN03658567	
*M. alpina **	CK1249	SAMN05720518	[27]
*M. alpina **	GBA31	SAMN05720773	[27]
*M. alpina **	LL118	SAMN20056918	[28]
*M. alpina **	NRRL 66262	SAMN10361219	[27]
*M. amoeboidea **	CBS 889.72	SAMN19911466	[31]
*M. antarctica **	KOD1030	SAMN05720520	[27]
*M. elongata **	AG-77	SAMN02745706	[30]
*M. epicladia **	AD058	SAMN05720441	[27]
*M. gamsii **	NVP60	SAMN05720530	[27]
*M. verticillata **	NRRL 6337	SAMN00699802	[29]
** *M. schmuckeri ** **	**CGMCC 20261**	**SAMN29492047**	**This study**
*Mucor circinelloides*	1006PhL	SAMN00103456	[60]
*M. lusitanicus*	CBS 277.49	SAMN00120579	[61]
*Parasitella parasitica*	CBS 412.66	SAMEA278055	[56]
*Rhizopus arrhizus*	GL8	SAMN14162349	[62]
*R. microsporus*	ATCC 52813	SAMN06821222	[63]
*Phycomyces blakesleeanus*	NRRL 1555	SAMN00189023	[61]
*Smittium culicis*	GSMNP	SAMN04489870	[64]
*Syncephalastrum monosporum*	B8922	SAMN02370995	
*Thermomucor indicae-seudaticae*	HACC 243	SAMN03070115	
*Umbelopsis isabellina*	WA0000067209	SAMN16393839	[65]
*U. vinacea*	WA0000051536	SAMN16393840	

Note: The star mark “*” represents members in the *Mortierellaceae*. The two genomes newly generated in this study are in bold.

**Table 2 jof-08-00891-t002:** Genomic features of *Mortierella* sequenced and de novo assembled in this study.

Species		*M. alpina* CGMCC 20262	*M. schmuckeri* CGMCC 20261
Genome size (Mb)		40.22	49.24
Scaffolds		28	25
Largest scaffolds (Mb)		4.35	4.78
GC (%)		50.92	47.46
N50 (Mb)		2.53	2.71
L50		6	8
Assembly BUSCO coverage (%)		97.4	97.6
PCG models		11,761	13,051
	Pfam	8507	8636
	NT	4026	4434
	NR	4904	6180
	GO	5411	5571
	Eggnog	9076	9436
	KEGG	4858	4921
	CAZymes	229	222
Gene clusters of secondary metabolites			
	Terpene	4	2
	Fungal-RiPP	0	1
	NRPS	15	0
	NRPS-like	1	2
	Siderophore	2	0
Repetitive elements (% in genomes)		7.29	7.45
ncRNA			
	rRNA	38	29
	tRNA	226	262

## Data Availability

The newly identified genomics sequences have been deposited in GenBank (Table 2).

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
