# Peer review of "Comparative Genomics of Mortierellaceae Provides Insights into Lipid Metabolism: Two Novel Types of Fatty Acid Synthase"

_jof, 2022, doi:10.3390/jof8090891_

Round 1
Reviewer 1 Report
Title: attractive and the subject is associated with a pivotal part of sustainable development axes.
Abstract: informative, but it is better to highlight the part of conclusion in the end of abstract.
Line 14#, in scientific or academic format it is better to replace eye-catching with other synonyms such outstanding, notable, remarkable, etc.
Lines 30-32# this paragraph needs to be more concise and the authors could replace "on" by "related to" or "involved in"
Introduction: authors should clarify why they intend the studying of Mortierellaceae rather than other relative Zygomycetes and other yeasts. Effect of growth rate, nutritional, physicochemical factors (Temperature and pH for example) should be mentioned as well. So, I suggest some articles helping you to import this part into introduction.
Consolidated Bioprocessing of Sugarcane Bagasse to Microbial Oil by Newly Isolated Oleaginous Fungus: Mortierella wolfii | SpringerLink
Isolation, identification, and statistical optimization of a psychrotolerant Mucor racemosus for sustainable lipid production | SpringerLink
Eco-Green Conversion of Watermelon Peels to Single Cell Oils Using a Unique Oleaginous Fungus: Lichtheimia corymbifera AH13 | SpringerLink
Sustainable lipid production from oleaginous fungus Syncephalastrum racemosum using synthetic and watermelon peel waste media - ScienceDirect
Characterization of Cellulase from Geotrichum candidum Strain Gad1 Approaching Bioethanol Production | SpringerLink
Author should avoid excessive citations (only 5 paragraphs in the introduction section with 26 references, I think it is too much).
Materials and methods:
Line 75#: both fungal strains were provided by China General Microbiological Culture Collection Center, Beijing, China (CGMCC), and there is no need to mention the previous isolation site to avoid illustration of full identification strategy.
Line 86#: the production conditions of lipid is totally missed; authors should mention the incubation conditions at which the fungus/fungi could synthesize the lipids
Line 99#: please clarify the incubation conditions if you followed static or shaking conditions. Also, clarify why you use 20 C as incubation temperature (I think the optimum temperature is 25 C, so why??).
Results
The outstanding part of the paper, it was carefully written, but if available to design a figure clarifying the genomic differences between both fungal strains rather than figure (3).
Discussion: good but I notices excessive citations, please minimize.
Conclusion: this part in unclear, please carefully write this part to highlight your findings and the research outcomes.
Update your old references if you can.
Author Response
Dear Mr. Hank Zhang and reviewers,
Thanks for your comments! We have carefully considered all the suggestions and tried our best to improve this manuscript. Point-to-point answer is as follows.
Title: attractive and the subject is associated with a pivotal part of sustainable development axes.
Response: Thank you!
Abstract: informative, but it is better to highlight the part of conclusion in the end of abstract.
Response: Thank you! We have revised this part as “This study provides novel insights into the enzymes related to the lipid metabolism, especially the ARA-related Delta5, ELOVL3 and FAS, laying a foundation for genetic engineering of Mortierellaceae to modulate yield in polyunsaturated fatty acids.”
Line 14#, in scientific or academic format it is better to replace eye-catching with other synonyms such outstanding, notable, remarkable, etc.
Response: Thanks! We used the "remarkable" to replace "eye-catching".
Lines 30-32# this paragraph needs to be more concise and the authors could replace "on" by "related to" or "involved in"
Response: Thank you! We are used the "related to" to replace "on".
Introduction: authors should clarify why they intend the studying of Mortierellaceae rather than other relative Zygomycetes and other yeasts. Effect of growth rate, nutritional, physicochemical factors (Temperature and pH for example) should be mentioned as well. So, I suggest some articles helping you to import this part into introduction.
Consolidated Bioprocessing of Sugarcane Bagasse to Microbial Oil by Newly Isolated Oleaginous Fungus: Mortierella wolfii | SpringerLink
Isolation, identification, and statistical optimization of a psychrotolerant Mucor racemosus for sustainable lipid production | SpringerLink
Eco-Green Conversion of Watermelon Peels to Single Cell Oils Using a Unique Oleaginous Fungus: Lichtheimia corymbifera AH13 | SpringerLink
Sustainable lipid production from oleaginous fungus Syncephalastrum racemosum using synthetic and watermelon peel waste media - ScienceDirect
Characterization of Cellulase from Geotrichum candidum Strain Gad1 Approaching Bioethanol Production | SpringerLink
Author should avoid excessive citations (only 5 paragraphs in the introduction section with 26 references, I think it is too much).
Response: Thanks for your suggestion. In Introduction section, we added some background and references about oleaginous fungi according to the list you provided. We believe these references will help demonstrate the importance of our study and attract more public attention.
"Therefore, oleaginous fungi such as Aspergillus flavus Link [=A. oryzae (Ahlb.) Cohn], Mortierella alpina Peyronel, M. wolfii B.S. Mehrotra & Baijal, Mucor racemosus Fresen., Lichtheimia corymbifera (Cohn) Vuill., Syncephalastrum racemosum Cohn ex J. Schröt., Geotrichum candidum Link, Umbelopsis isabellina (Oudem.) W. Gams (=Mortierella isabellina Oudem.) and Yarrowia lipolytica (Wick., Kurtzman & Herman) Van der Walt & Arx, have recently gained attention and played an important role in the studies of bio-diesel and polyunsaturated fatty acids (PUFAs) [1–11]."
"Compared with other microorganisms like microalgae and algae, oleaginous fungi are distinct in possessing a shorter life cycle, assimilating a variety of carbon sources, and adapting to various weather and seasons [1, 7, 9]."
Materials and methods:
Line 75#: both fungal strains were provided by China General Microbiological Culture Collection Center, Beijing, China (CGMCC), and there is no need to mention the previous isolation site to avoid illustration of full identification strategy.
Response: Thanks! The isolation site deleted in this MS.
Line 86#: the production conditions of lipid is totally missed; authors should mention the incubation conditions at which the fungus/fungi could synthesize the lipids
Response: Thank you! The incubation conditions is demonstrated as "1 mL of spore suspension (1 × 106) was incubated at 20 °C and 140 rpm for seven days in a 250 mL flask with 100 mL of modified Kendrick media" in the section "2.1. Strains, media and fermentation".
Line 99#: please clarify the incubation conditions if you followed static or shaking conditions. Also, clarify why you use 20 C as incubation temperature (I think the optimum temperature is 25 C, so why??).
Response: Thanks! Yes, the optimum temperature is 25 C in many strains of Mortierella, while the Mortierella alpina synthesized more ARA in 20 C based on our pre-experiments.
Results
The outstanding part of the paper, it was carefully written, but if available to design a figure clarifying the genomic differences between both fungal strains rather than figure (3).
Response: Thanks! However, we couldn’t find a more desirable figure to show their genomic differences than Table 2 displaying the genomic differences of these two strains. In order to provide an intuitive comparison, we shew the genomic synteny of Mortierella alpina CGMCC 20262 and M. schmuckeri CGMCC 20261.
Discussion: good but I notices excessive citations, please minimize.
Response: Thank you!
Conclusion: this part in unclear, please carefully write this part to highlight your findings and the research outcomes.
Response: Thanks for your suggestion, we added the conclusion in this study as follows:
"In this paper, two new genomes, Mortierella alpina CGMCC 20262 and M. schmuckeri CGMCC 20261, were sequenced using the PacBio Sequel and Illumina NovaSeq 6000 platform. To explore the lipid metabolism in Mortierellaceae, a total of 19 genomes were reannotated. The results suggest that delta-5 desaturase and elongation of very long chain fatty acids protein 3 probably promoted the accumulation of polyunsaturated fatty acids, especially arachidonic acid. Besides, three types of fatty acid synthase including two novel ones, were identified. Consequently, with the increase of public genomic data, a comprehensive analysis on lipid metabolism will find out more genes or protein-encoding models, providing more information for subsequent genetic engineering of oleaginous fungi"
Update your old references if you can.
Response: Yes, the references have been updated.
Reviewer 2 Report
Dear Editors and Authors,
The author reports two novel and high-quality genomes of Mortierella alpina CGMCC 20262 and M. schmuckeri CGMCC 20261. This study provides novel insights into the enzymes on the lipid metabolism, laying a foundation for genetic engineering of Mortierellaceae to modulate yield in polyunsaturated fatty acids. The results from the study of the lipid mechanism of the Mortierellaceae family supports the fact that the rapid increase in genomic data provides more materials for tackling fatty acid biosynthesis, discovering more potential genes, and finalizing the understanding of the genetic foundation of lipid metabolism.
Excellent, robust work. Hard to find anything to improve. The topic is interesting, and the results and graphs are presented sophisticatedly. Perhaps a sentence or two could be used to outline future goals, as it would be interesting to see what the authors' further aims are.

Author Response
Dear Mr. Hank Zhang and reviewers,
Thanks for your comments! We have carefully considered all the suggestions and tried our best to improve this manuscript. Point-to-point answer is as follows.
The author reports two novel and high-quality genomes of Mortierella alpina CGMCC 20262 and M. schmuckeri CGMCC 20261. This study provides novel insights into the enzymes on the lipid metabolism, laying a foundation for genetic engineering of Mortierellaceae to modulate yield in polyunsaturated fatty acids. The results from the study of the lipid mechanism of the Mortierellaceae family supports the fact that the rapid increase in genomic data provides more materials for tackling fatty acid biosynthesis, discovering more potential genes, and finalizing the understanding of the genetic foundation of lipid metabolism.
Excellent, robust work. Hard to find anything to improve. The topic is interesting, and the results and graphs are presented sophisticatedly. Perhaps a sentence or two could be used to outline future goals, as it would be interesting to see what the authors' further aims are.
Response: Thanks for your encouragement! We added a conclusion in this study.
"In this paper, two new genomes, Mortierella alpina CGMCC 20262 and M. schmuckeri CGMCC 20261, were sequenced using the PacBio Sequel and Illumina NovaSeq 6000 platform. To explore the lipid metabolism in Mortierellaceae, a total of 19 genomes were reannotated. The results suggest that delta-5 desaturase and elongation of very long chain fatty acids protein 3 probably promoted the accumulation of polyunsaturated fatty acids, especially arachidonic acid. Besides, three types of fatty acid synthase including two novel ones, were identified. Consequently, with the increase of public genomic data, a comprehensive analysis on lipid metabolism will find out more genes or protein-encoding models, providing more information for subsequent genetic engineering of oleaginous fungi."
Reviewer 3 Report
Polyunsaturated fatty acids (PUFAs) are increasingly in demand due to the beneficial effects on human healthArachidonic acid is one of the main PUFAs produced from M. alpina. As one of the oleaginous fungi, however, M. alpina can also produce other PUFAs in significant amounts, such as LA, GLA, DGLA , EPA, and DHA, which might be induced by lowering the temperature or adding exogenous oils, suggesting this fungus is a potential producer of many commercially important PUFAs, both n-3 and n-6. In addition, as the fungus could produce AA and DHA, oil products with good n-3 and n-6 ratios may be developed as new desired food products. I have few comments to improve this article:
How this publication has novelty over your others research on similar topic?
Betina et al., 1980 used chloroform and methanol for lipid extraction so why authors used different methods for lipid extraction as per current knowledge HCL treatment can break the fatty acid chain.
Rtx-5MS, RESTEK is not good column to analyse the FAME, its mainly for hydrocabons semivolatiles, PAHs, chlorinated hydrocarbons, phthalates, phenols, amines, organochlorine and organophosphorus pesticides, drugs and solvent impurities.
Figure 1 is bit confusing without proper axis title and total lipid content unit. The data should be provided as g/L not as g/100ml. Fatty acid profile should be presented as table in triplicate.
Author should try to charaterize the desaturases by mutating these two micrbes rather than providing only examples of other.
How did you draw Fig 4 and 5, please provide sutaible refences for the same.
Author Response
Dear Mr. Hank Zhang and reviewers,
Thanks for your comments! We have carefully considered all the suggestions and tried our best to improve this manuscript. Point-to-point answer is as follows.
Polyunsaturated fatty acids (PUFAs) are increasingly in demand due to the beneficial effects on human health Arachidonic acid is one of the main PUFAs produced from M. alpina. As one of the oleaginous fungi, however, M. alpina can also produce other PUFAs in significant amounts, such as LA, GLA, DGLA , EPA, and DHA, which might be induced by lowering the temperature or adding exogenous oils, suggesting this fungus is a potential producer of many commercially important PUFAs, both n-3 and n-6. In addition, as the fungus could produce AA and DHA, oil products with good n-3 and n-6 ratios may be developed as new desired food products. I have few comments to improve this article:
How this publication has novelty over your others research on similar topic?
Response: Thanks for your question. Firstly, in this paper, two new genomes, Mortierella alpina CGMCC 20262 and M. schmuckeriCGMCC 20261, were sequenced using the PacBio Sequel and Illumina NovaSeq 6000 platform. Secondly, a total of 19 genomes of Mortierellaceae supports the fact that the increasing genomic data provides more materials for tackling fatty acid metabolism, discovering more potentials genes. Finally, two novelty fatty acid synthase are identified.
Betina et al., 1980 used chloroform and methanol for lipid extraction so why authors used different methods for lipid extraction as per current knowledge HCL treatment can break the fatty acid chain.
Response: Thanks! There are two reasons for not using chloroform in this study. First, the laboratory has strict requirements on the use of chloroform, which makes it difficult to purchase and use. Secondly, for the safety of experimenters. Therefore, the HCL became an alternative.
Rtx-5MS, RESTEK is not good column to analyses the FAME, its mainly for hydrocarbons semivolatiles, PAHs, chlorinated hydrocarbons, phthalates, phenols, amines, organochlorine and organophosphorus pesticides, drugs and solvent impurities.
Response: Thanks for your suggestion! we will use a more suitable column such as TG–5MS (30 m × 0.25 mm × 0.25 μm film thickness) in the future studies, and we would appreciate if you have a recommendation.
Figure 1 is bit confusing without proper axis title and total lipid content unit. The data should be provided as g/L not as g/100ml. Fatty acid profile should be presented as table in triplicate.
Response: Thanks for your suggestion. We used the "g/L" to replace the "g/100ml" in the figure 1a and manuscript. However, the fatty acid profile presented through the figure 1b is more intuitive than through a table for showing the huge differences between Mortierella alpinaCGMCC 20262 and M. schmuckeri CGMCC 20261.
Author should try to charaterize the desaturases by mutating these two microbes rather than providing only examples of other.
Response: Thanks for your suggestion! Our team has not established a related genetic manipulation system for Mortierella alpina, although we are planning to do so.
How did you draw Fig 4 and 5, please provide suitable refences for the same.
Response: Thanks! The figure 4 draw using the Adobe Photoshop based on KEEG pathway. For figure 5, the protein sequences of fatty acid synthase were obtained (Supplementary File S1), then Blastp was used to annotate the active site, and finally the different active sites were summarized and visualized using Adobe Photoshop.
Round 2
Reviewer 3 Report
The authors responded to all queries satisfactorily, so I can recommend for publication.